# Effects of Different Energy Intensities of Microwave Treatment on Heartwood and Sapwood Microstructures in Norway Spruce

Sauradipta Ganguly [1], Angela Balzano [2], Marko Petrič [2], Davor Kržišnik [2], Sadhna Tripathi [1], Jure Žigon [2] and Maks Merela [2,*]

1 Wood Preservation Discipline, Forest Products Division, Forest Research Institute, Dehradun 248006, India; sauradipta.ganguly@fridu.edu.in (S.G.); tripathis@icfre.org (S.T.)
2 Department of Wood Science and Technology, Biotechnical Faculty, University of Ljubljana, 1000 Ljubljana, Slovenia; angela.balzano@bf.uni-lj.si (A.B.); marko.petric@bf.uni-lj.si (M.P.); davor.krzisnik@bf.uni-lj.si (D.K.); jure.zigon@bf.uni-lj.si (J.Ž.)
* Correspondence: maks.merela@bf.uni-lj.si; Tel.: +386-1-320-3646

**Abstract:** Microwave modification can increase the permeability of wood by delaminating and rupturing its anatomical microstructures at their weak points. A high degree of intensity of microwave modification can cause significant structural damage to the microstructures of wood, resulting in poorer strength properties. The objective of this study was to evaluate the changes in the anatomical structure of Norway spruce (*Picea abies* (L.) Karst.) heartwood and sapwood after microwave modification in order to develop the most effective treatment in terms of applied energy without causing significant structural damage. Analysis with light and scanning electron microscopy were performed to evaluate the effect of microwave treatment for two different energy intensities, moderate and high intensity. The results indicated structural changes in the tracheid cells. Microscopy showed varying degrees of modification within the wood microstructure, with the heartwood samples showing a greater anatomical distortion compared to their sapwood counterparts. Furthermore, the samples were subjected to pycnometric density measurements, which indicated a reduction in skeletal and absolute density after microwave modification, for both high and moderate intensity treatment on sapwood and heartwood samples. With increasing microwave energy, a gradual increase in specific pore volume and porosity percentage of the samples were also detected.

**Keywords:** *Picea abies*; microwave treatment; light microscopy; scanning electron microscopy; wood anatomy; wood properties; density; porosity; permeability



## 1. Introduction

Wood is used in various sectors due to its environmentally friendly nature, standard mechanical properties, machinability, natural origin, ability to sequester carbon in service and aesthetic appearance, and these properties make it an excellent natural building material. The diversity of woody biomass available worldwide almost guarantees the use of different types of wood species with the required properties for specific end-uses [1]. However, the increasing global demand for high-quality durable material coupled with depleting reserves of wood species with inherent natural durability has led to a shift in emphasis towards some plantation wood species whose wood is not durable. Moreover, despite its undeniable superiority over other building materials, the use of wood is often limited due to its inherent properties such as biological decay, poor dimensional stability, poor treatability, and high fire susceptibility, which are highly undesirable in service [2,3]. For example, wood species with low permeability are associated with many difficulties in impregnation with preservatives and resins. Low permeability also causes problems associated with wood-drying processes, which can be time consuming, expensive, and even ineffective.

Therefore, recent research in the niche sector of wood science and technology has focused on various wood modification processes to overcome the shortcomings associated with processing and use in service [4]. Thermal and chemical modifications of wood are the most common in practice today and have found commercial applications after satisfactory performances in terms of improved biological resistance and dimensional stability [5–14] but are often associated with high energy consumption and the use of chemicals that are not environmentally friendly.

Microwave (MW) wood modification is an environmentally friendly modification technique with relatively low energy consumption [15] when compared to some other modification techniques, and supports various wood processing operations by significantly improving permeability, treatability, and uptake of wood preservatives [16–22]. Microwave treatment of wood is particularly useful for less permeable wood, as in this way we can improve the impregnability of the wood and achieve more effective protection to increase its resistance. Depending on the MW energy used, a several thousand-fold increase in wood permeability in radial and longitudinal directions can be achieved [16,23–25]. The increase in permeability and fluid flow within the wood microstructure is mainly achieved by rupturing the weak anatomical structures such as ray cells and parenchyma [26,27], softening and mobilizing resin, and forming a large number of voids in the radial longitudinal directions [24]. The applied MW energy leads to heating of the water molecules in the wood to boiling point and then generates a high vapor pressure within the wood cells. The pressure gradient increases within 0.3–6 s after the applied MW energy and the vapor pressure can reach up to 6 bar [16], leading to fragmentation of weak anatomical structures in the wood and often flushing out vascular deposits such as tyloses and reducing the degree of occlusion.

The change in anatomical structure is often related to the energy intensity of the MW treatment which, together with the time, is a determining factor in the level of structural changes or even damage [28]. However, to the best of our knowledge the availability of similar literature is not so frequent. Cell wall delamination, formation of micro-cracks and partial or complete destruction of pit membranes were reported [27] for Chinese fir (*Cunninghamia lanceolata* (Lamb.) Hook) after MW treatment. Similar findings and further significant improvement in average pore diameter after MW treatment of Scots pine (*Pinus sylvestris* L.) was reported [29], which agrees with previous findings for larch wood [25]. The MW treatment and its effects on the anatomical properties of wood were recently studied in two species, where the effect of MW treatment was more pronounced in Radiata Pine (*Pinus radiate* D. Don) than in Norway Spruce (*Picea abies* (L.) Karst.) and the authors reported the frequent occurrence of modified pit aperture and major changes in the S3 and warty layers, which were evident after higher degrees of modification [30].The most frequent pattern of checking was found along the ray cells in both species, with frequent cleavage of the middle lamella. In a similar study [31], it was reported that MW treatment resulted in the crack formation in the cross-field pit region, propagating to the nearby cell walls which increased in length depending on the intensity (cracks length reached 2–4 times of the long axis diameter of the pit). Distortion in the middle lamella between ray parenchyma cells and longitudinal tracheids were also reported, resulting in separation of the two components with crack width generally between 1 and 25 μm. Such anatomical changes after MW treatment are more pronounced at higher energy exposure and can lead to a reduction in strength properties depending on the species and type of wood [16]. Although, this strength reduction is often not significant [27] and is more pronounced on the tangential direction than on the radial side [16]. Furthermore, it has been emphasized that wood species with wide sapwood rings are not strongly affected by MW treatment, resulting in a small change in bending strength. However, the extent and execution of MW treatment varies from species to species and is highly dependent on the initial moisture content (IMC) of the wood, which may result in no significant structural damage caused to wood with better structural integrity when the low or moderate degree of MW treatment is carried out [28].

To assess the effect of MW treatment on the microstructure of wood, recently, scanning electron microscopy (SEM) has been widely used [30,31]. SEM provides high-resolution images at high magnification, making it ideal for studying the ultrastructure and topography of wood and the distribution of its features [32].

In the present study, we used light microscopy as well as SEM to evaluate the effect of MW treatment on the microstructure of Norway spruce, a wood species that is commercially important in Europe and widely distributed worldwide. We aimed to evaluate the effect on the microstructure of both heartwood and sapwood after two different energy intensities of MW treatment, corresponding to a moderate intensity (MI) and a high intensity (HI) degree of modification. It has been confirmed in Scots pine (*Pinus sylvestris* L.) that the mutual influence of anatomical and chemical characteristics as a result of heartwood formation affects anisotropy, swelling and sorption quotient [33], density and mechanical properties in different oak species [34] and, therefore, it is important to analyze both sapwood and heartwood.

In addition, absolute density measurements in helium and envelope density displacement measurements were performed to determine skeletal and envelope volume measurements. This allows the calculation of the percentage porosity and the total pore volume of the wood samples. In this way, we have tried to correlate the anatomical changes that take place after MW treatment with the change in density. The changes in wood density after MW treatment have not been extensively studied until recently and may therefore complement the results for a better understanding of the method.

The final objective of this study was to evaluate the changes in the heartwood and sapwood microstructure of Norway spruce wood to develop optimal and effective MW treatment, in terms of applied energy without causing significant wood structural damage.

## 2. Materials and Methods

### 2.1. Materials

Wood of Norway spruce, 35 years old, cut in the north-eastern part of Slovenia, was considered for the study. The logs were processed into planks at a local sawmill. Samples were mechanically processed (sawing, planing) three weeks after cutting the tree. Separate sets of samples, mostly semi-radial in orientation, were prepared from sapwood and heartwood. The initial moisture content (IMC) was determined by the gravimetric method for 10 randomly selected parallel samples and for the sapwood samples it was 90–120%, while the IMC of heartwood samples was 27–38%. After final dimension preparation (l × w × h = 5 × 2.5 × 1.5 cm$^3$) for MW treatment and (l × w × h = 1.0 × 1.0 × 1.0 cm$^3$) for pycnometric analyses the samples were conditioned at 60% air relative humidity (RH) and 20 °C until constant mass. This was necessary before exposure to MW radiation to ensure uniform moisture content and distribution in each sample set.

### 2.2. Microwave Treatment

The treatment of wood with microwaves (MW) was carried out using a MW oven (Model: M020MW, Gorenje, Velenje, Slovenia) with a frequency of 2.45 GHz and a maximum output of 700 W. Two MW energies of 700 MJ/m$^3$ and 1260 MJ/m$^3$, corresponding to moderate (MI) and high (HI) intensities, were applied to the wood samples. The duration of MW irradiation was 20 and 36 s respectively for MI and HI. Parameters of MW treatments were defined based on MW power and volume of the samples as proposed by Kol and Çayir [35] and our preliminary experiments. After MW treatment, the samples were cooled in a desiccator at room temperature and subjected to further analysis.

### 2.3. Pycnometric Density Measurements

The gas displacement pycnometer AccuPyc II 1340 (Micromeritics Inc., Norcross, GA, USA) was used to measure skeletal volume and skeletal density by measuring the pressure change of helium in calibrated volumes. The samples were dried in an oven at 60 °C for 72 h and then at 103 °C for 24 h before the experiment until a constant mass

was achieved. Nine samples were measured for each combination of treatment and wood category (sapwood and heartwood). A total of 27 sapwood and 27 heartwood samples were measured (Table 1). Each set of nine samples was weighed in the oven-dry condition and loaded into the 35 cm$^3$ cell chamber. The chamber insert containing the samples was purged 10 times to clean the samples and remove air and moisture from inside the chamber. For each set, volume and specific gravity were measured five times until an equilibrium rate of 0.020 psig/min was reached; only average values are considered for this manuscript. The skeletal density (also called absolute, true, real and apparent density) is obtained when the measured volume excludes the pores within the bulk sample. The envelope density (also called bulk density) is determined for porous materials when the pore spaces within the material are included in the volume measurement.

**Table 1.** Pycnometric density measurements. Microwave treatment (MW) was performed at moderate (MI) and high (HI) intensities on the Sapwood and Heartwood of Norway spruce (*Picea abies*).

| | AccuPyc | | GeoPyc | | | |
|---|---|---|---|---|---|---|
| Sample | Skeletal Density (g/cm$^3$) | St. Dev. | Envelope Density (g/cm$^3$) | St. Dev | Specific Pore Volume (cm$^3$/g) | Porosity (%) |
| Control Sapwood | 1.4241 | 0.0025 | 0.5565 | 0.0008 | 1.0955 | 60.92 |
| MI MW Sapwood | 1.4178 | 0.0031 | 0.5324 | 0.0012 | 1.1510 | 61.88 |
| HI MW Sapwood | 1.3611 | 0.0006 | 0.4802 | 0.0012 | 1.3505 | 64.72 |
| Control Heartwood | 1.3577 | 0.0062 | 0.5176 | 0.0019 | 1.1950 | 61.28 |
| MI MW Heartwood | 1.3314 | 0.0039 | 0.4782 | 0.0010 | 1.3400 | 64.08 |
| HI MW Heartwood | 1.2180 | 0.0067 | 0.5085 | 0.0004 | 1.1450 | 59.25 |

Envelope density measurements were performed using the GeoPyc 1365 (Micromeritics Inc., Norcross, GA, USA). The GeoPyc instrument uses a quasi-fluid displacement medium with high flowability that does not fill the pores of the samples. The samples were dried to constant mass in an oven at 60 °C for 72 h and then at 103 °C for 24 h. For each sample, the envelope volume was determined using a 19.1 mm diameter sample chamber at 38.0 N consolidation force and a conversion factor of 0.2907 cm$^3$/mm, as recommended in the instruction manual [36]. The sample chamber is initially filled with fine (fluid) sand only and packed under rotation until to the consolidation force while rotating. The volume of the sand is calculated from the position of the piston. Then the sample chamber is opened, and the weighed sample is transferred to the sample chamber. The recommended volume of the sample to the displacement medium (25 percent by volume) was followed. Consolidation by a specific force was repeated several times; 2 preparation cycles and 5 measuring cycles to obtain a uniform packing.

The combination of skeletal and envelope volume measurements allows the calculation of the percentage porosity, as in Equation (1):

$$\text{Percent porosity (\%)} = \frac{(\rho_{AccuPyc} - \rho_{GeoPyc})}{\rho_{AccuPyc}} \times 100 \tag{1}$$

where $\rho_{AccuPyc}$ is skeletal density achieved with AccuPyc, and $\rho_{GeoPyc}$ is envelope density achieved with GeoPyc. Specific pore volume was calculated as follows in Equation (2):

$$\text{Specific Pore Volume} \left( \frac{\text{cm}^3}{\text{g}} \right) = \frac{Percent\ porosity}{\rho_{GeoPyc}} \tag{2}$$

## 2.4. Light Microscopy and Scanning Electron Microscopy

Control and MW treated samples ($5 \times 2.5 \times 1.5$ cm$^3$) were cut from the middle and converted in small subsamples (cubes of 1 cm$^3$ dimension) for light microscopy and SEM.

For light microscopy, the subsamples were soaked in a mixture of distilled water, glycerol, and ethanol (60:35:5) for one week to soften the wood before cutting. Cross-sections (10 μm thick) were then cut using a sliding microtome LM2010R (Leica, Wetzlar, Germany) with a classical microtome blade (Leica), then sections were stained with a safranin (0.04%) and astra blue (0.15%) water solution [37–39] and mounted in Euparal (Bioquip Products Inc., Compton, CA, USA) [40]. Cross-sections were observed with a Nikon Eclipse 800 scientific light microscope (Nikon, Tokyo, Japan), and microphotographs were taken with a DS-Fi1 digital camera equipped with the NIS-Elements BR 3 image analysis system (Nikon, Melville, NY, USA).

The subsamples for SEM were prepared according to the methodology proposed by Merela and co-workers [32] to obtain an optimal surface for observation. From each MW treated sample, we prepared two oriented cubes (1 cm$^3$) (one for cross and one for radial xylotomic plane). To obtain an optimal surface for SEM observation, the considered anatomical plane of the subsample was moistened with a wet brush and then trimmed with a sliding microtome (Leica) with interchangeable low profile microtome blades (DB80 LX, Leica Biosystems, Nussloch, Germany) using a special blade holder—Low Profile Blade Rail Twin Set (Leica).

The samples were then dried at room temperature (22 °C) and at air RH of 65%, mounted on stubs with a conductive carbon adhesive strip and coated with an Au/Pd sputter coater (Q150R ES Coating System; Quorum technologies, Laughton, UK) for 60 s with a constant current of 20 mA. The SEM micrographs were taken at low voltage (5 kV) and in a low vacuum (50 Pa) using a large field detector (LFD) by a Quanta 250 scanning electron microscope (FEI Company, Hillsboro, OR, USA) at working distances between 7–10 mm. Anatomical changes (for instance cracks and cavities) were measured using ImageJ image analysis software [41]. Features from the SEM observations were used to discuss the effect of MW on the anatomical features.

## 2.5. Data Analysis

Statistical analyses were performed with the free software R (version 4.1.0) (R Core Team 2016, Vienna, Austria) using the packages tidyverse, ggpubr, and rstatix. Each observation is independent and there is no relationship between observations in each sample group. No significant outliers were observed. Normality was checked by analyzing the analysis of variance (ANOVA) model residuals in each sample group and calculating the Shapiro–Wilk test for each group. The homogeneity of variances was checked using Levene's test. If the ANOVA assumptions were confirmed, a one-sided ANOVA test was performed, followed by Tukey post-hoc tests to perform multiple pairwise comparisons between samples. For non-parametric data, a Kruskal–Wallis rank-sum test with a significance level of 0.05 was used to determine if there were systematic differences in scores between groups. Wilcoxon rank-sum test with continuity correction was performed to assess the strength and direction of ordinal association between samples.

## 3. Results and Discussion

### 3.1. Pycnometric Density

The oven-dry specific density of the cell wall substance was determined. The skeletal density and the envelope density were determined, and results are shown in Table 1 and Figure 1. The skeletal density of the sapwood samples was higher than that of the heartwood, with the highest density measured in the untreated sapwood samples (1.4241 g/cm$^3$). The skeletal density decreased with increasing intensity exposure to MW treatment. At MI, the skeletal density for sapwood samples was 0.4% lower compared to the control and 4.4% lower at HI. A higher decrease in skeletal density was observed for heartwood samples: 1.9% decrease at MI samples and 10.3% decrease at HI compared to

control heartwood samples. A similar effect could also be observed in the envelope density: 4.3% decrease in envelope density at MI and 13.7% at HI compared to control sapwood samples and 7.6% decrease at MI and 1.8% at HI compared to control heartwood samples. It appears that the MW treatment had a greater effect on the heartwood samples, excluding high intensity MW treatment on envelope density. The HI MW treated heartwood set behaved differently and did not follow the trend. Although the envelope density of this set was lower than that of the controls, it was surprisingly higher than that of the MI MW treated heartwood. This could perhaps be attributed to the higher proportion of latewood in these samples, what made them denser. Another reason may be the presence of wide rings with a large portion of earlywood in these samples (Figure 1), which may have negated the influence of MW treatment [16], resulting in a smaller reduction in density. In addition to this, mild deformation in the appearance of the samples can also be seen in some samples (Figure 1) from square cross section to rhomboid which could be a possible explanation.

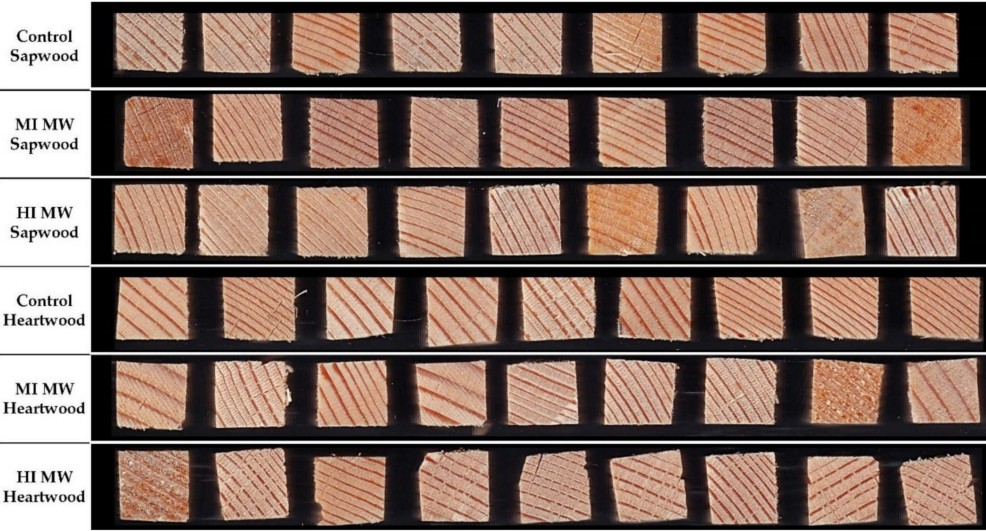

**Figure 1.** Samples ($1.0 \times 1.0$ cm$^2$) used for pycnometric density measurements. Microwave treatment (MW) was performed at moderate (MI) and high (HI) intensities on the Sapwood and Heartwood of Norway spruce (*Picea abies*).

The specific pore volume and percentage of porosity increased with increasing MW treatment, again excluding the HI MW treated heartwood (Table 1).

The skeletal density measured with AccuPyc met the ANOVA assumptions and consequently a one-way ANOVA was performed. One-way ANOVA showed that there were significant density differences between samples ($p$-value = $1.34 \times 10^{-26}$). The results of the Tukey post-hoc test proved the significant differences between sample groups ($p < 0.001$). There was no significant difference between the groups Control Sapwood and MI MW Sapwood ($p$-value = $4.82 \times 10^{-1}$).

The envelope density measured by GeoPyc did not meet the ANOVA assumptions, so a Kruskal–Wallis rank sum test was performed at a significance level of 0.05 and showed that there were significant differences between the samples ($p$-value = $2.2 \times 10^{-16}$). The results of the Wilcoxon rank sum test proved the significant differences between the sample groups ($p < 0.001$). There was no significant difference between the groups Control Heartwood and HI MW Heartwood ($p$-value = $5.60 \times 10^{-4}$).

Porosity measurements met the ANOVA assumptions and a one-way ANOVA was performed, which showed significant differences between samples ($p$-value = $2.78 \times 10^{-7}$). In addition, the results of the Tukey post-hoc test proved the significant differences between the sample groups ($p < 0.001$). We found no significant difference in porosity between Control Sapwood and MI MW Sapwood ($p$-value = $6.27 \times 10^{-1}$) and MI MW Sapwood and HI MW Sapwood ($p$-value = $3.39 \times 10^{-1}$).

If we assume that porosity increase is caused by structural changes in wood anatomy such as collapse and delamination of the cell wall in both earlywood and latewood, we can hypothesize that HI MW treatment should be avoided in heartwood, while the HI MW treatment does not cause structural damage in the sapwood. On the other hand, the decrease in skeletal density with the increase in MW treatment intensity could be related to the extractive migration to the cell wall surface and the consequent apparent increase in the volume of cell wall substance.

### 3.2. Light Microscopy and Scanning Electron Microscopy of Microwave-Treated Samples

Light microscopy observation revealed significant differences between the control (Figure 2A) and the MW-treated samples (Figure 2C,E,G,I). There was a general change in the tracheid cell wall structure. This result agrees with previous studies describing the effect of different intensity degrees of MW treatment. It has been reported that microwave treatment can cause depolymerisation of lignin [42], decomposition and partial destruction of chemical bonds between cellulose and lignin and between cellulose and hemicelluloses [43]. An advantage of using light microscopy, in this case, is the possibility to detect chemical changes in the main structural polymers of wood cell walls by different tissue staining. However, the change in the wood structure after treatment made it difficult to obtain thin cross-sections needed for light microscopy. After MW treatment, the structure of the wood became more brittle and when cutting with a microtome, thin cross-section wood slices were partly damaged. Although light microscopy provides a good overview of tissue changes, we needed to examine the changes at the microstructural level with higher resolution and without the damage caused by sample preparation that occurs with the light microscopy method. For this reason, we performed SEM investigation in the anatomical structure at the cellular level for each sample series.

SEM examination of the control samples (Figure 3A,B) showed that the intact microstructure of the wood before treatment was preserved. In the cross-section of both earlywood (Figure 3A,C) and latewood (Figure 3B,D), the tracheid wall structure appeared undamaged, with no detachment of the middle lamella regions between tracheids. In the radial section, we observed a longitudinal tracheid with a normal intercellular layer (Figure 3E,F) with many bordered pits with intact pit membrane (Figure 3E,G) and undamaged cross-field pits (Figure 3F,H) on the tracheid walls.

Compared to the control we observed typical delamination between cell walls on the cross-section of the treated samples at both MI (Figure 2D,F) and HI (Figure 2H,J) MW treatment. SEM enabled detection of distortions in the earlywood tissue structure in treated heartwood samples, especially at the HI MW treatment level. Detailed SEM analyses for each material and treatment are reported in the following sections.

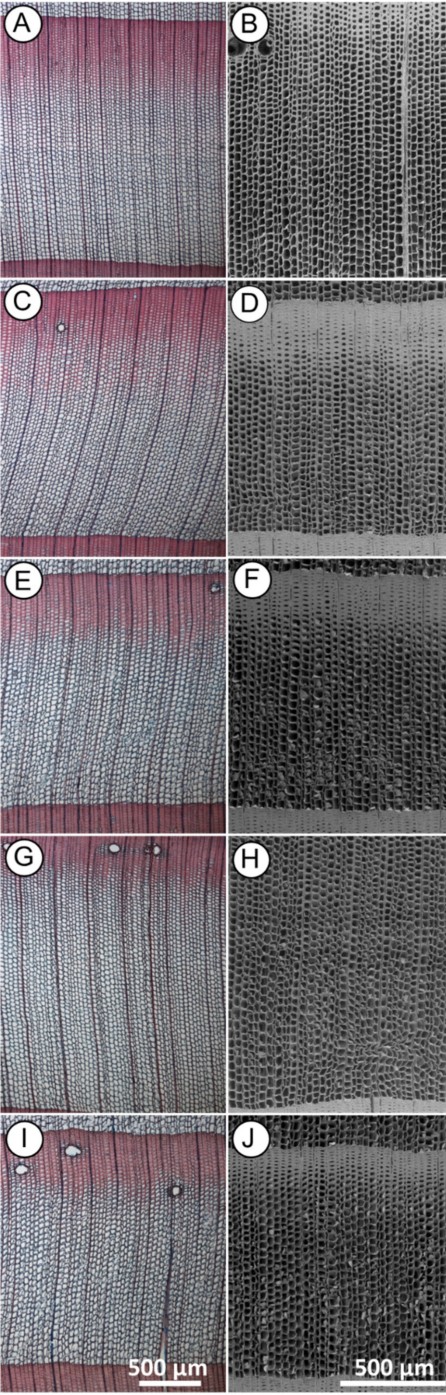

**Figure 2.** Cross-sections of Norway spruce observed with light microscopy (**A**,**C**,**E**,**G**,**I**) and scanning electron microscopy (SEM) (**B**,**D**,**F**,**H**,**J**) showing general changes in the tracheid cell walls structure between control and after microwave (MW) treatment. ((**A**,**B**)—untreated wood; (**C**,**D**) heartwood and (**E**,**F**) sapwood)—MW treated with moderate intensity (MI); ((**G**,**H**) heartwood and (**I**,**J**) sapwood)—MW treated with high intensity (HI). Scale bar = 500 μm.

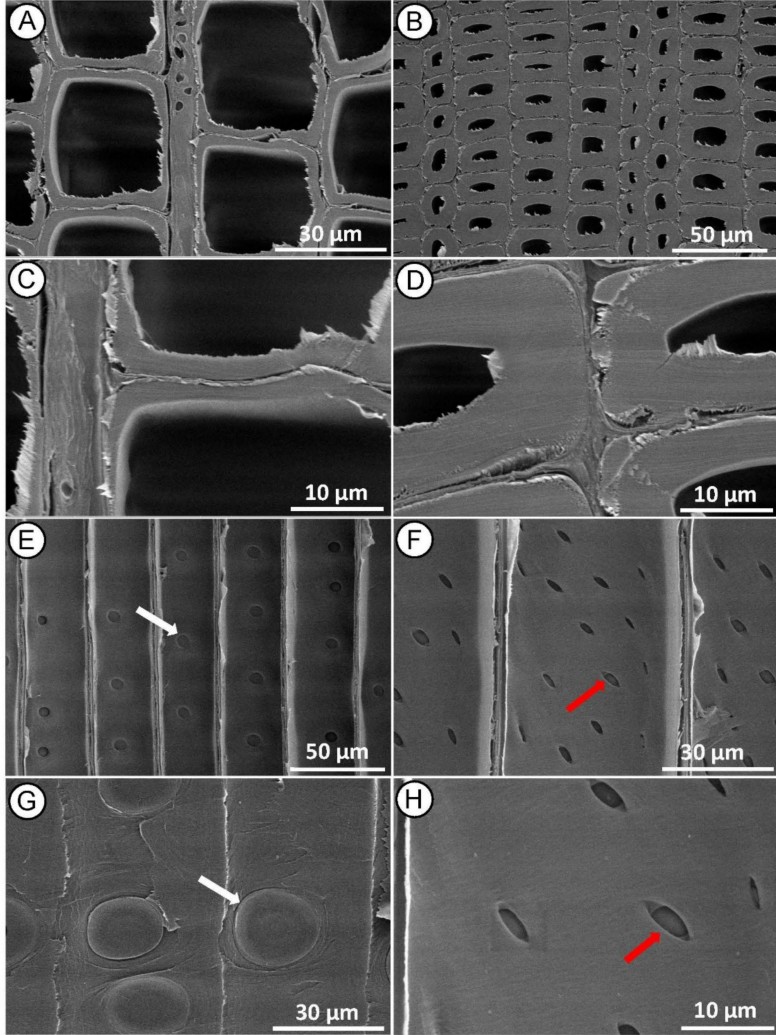

**Figure 3.** SEM micrographs of control (untreated) Norway spruce. Cross-section showing normal tracheid structure in earlywood (**A,C**) and latewood (**B,D**). Radial section with intact bordered pits on the tracheid walls ((**E,G**)—white arrows) and undamaged-intact piceoid cross-field pits ((**F,H**)—red arrows).

### 3.3. Moderate Intensity Microwave Treated Heartwood and Sapwood

Microscopy analyses of MI MW treated heartwood on cross section showed distortions in the earlywood tissue (Figure 4A), collapse and delamination of the cell wall, both in the earlywood (Figure 4A,C) and latewood (Figure 4B,D). Same images revealed that many microcracks were formed along the radial and tangential walls of tracheids between middle lamella and the primary wall. The middle lamella is indeed a weak point in the wood structure, and it is the first layer to be damaged by the high-pressure steam produced during the MW treatment. Fewer cracks also occurred across the S2 layer and warty layer (Figure 4D), as previously reported in Radiata pine [30].

Radial section (Figure 4E–G) revealed longitudinal delamination of middle lamella between adjacent tracheids, but no significant changes in other wood microstructures were found. Bordered pits on tracheid walls appeared with a normal structure, mostly aspirated, and only a few of them with a de-aspirated membrane [44] (Figure 4E,G). The cross-field pits also appeared mostly intact, rarely with small cracks (Figure 4F,H), indicating that the treatment conditions were not sufficient to induce large structural changes in these anatomical structures. Changes in the microstructure of the wood, as described in this case, may not significantly affect its mechanical properties, but may facilitate impregnation treatment because the wood preservative flows more easily through the cell wall [45–47].

In the sapwood samples, the earlywood tissue observed in the cross-section was not distorted as in the case of the heartwood, although it also showed signs of cell wall collapse and delamination (Figure 5A,C), while the latewood structure appeared largely normal (Figure 5B,D). In contrast to the heartwood, the greatest damage in the sapwood was observed in the radial section, where longitudinal delamination of the middle lamella was evident (Figure 5E,F), the bordered pit membranes were damaged (cracked) (Figure 5E,G) and the cross-field pits apertures reveal micro crack between 0.5 and 2 µm at the edges (Figure 5F,H).

Cracking of cross-field pits poles after MW treatment has also been reported in Chinese fir heartwood [31] and Norway spruce [45] because of a steam explosion during high-frequency drying, showing that cross-field pits were easily damaged by high-pressure steam.

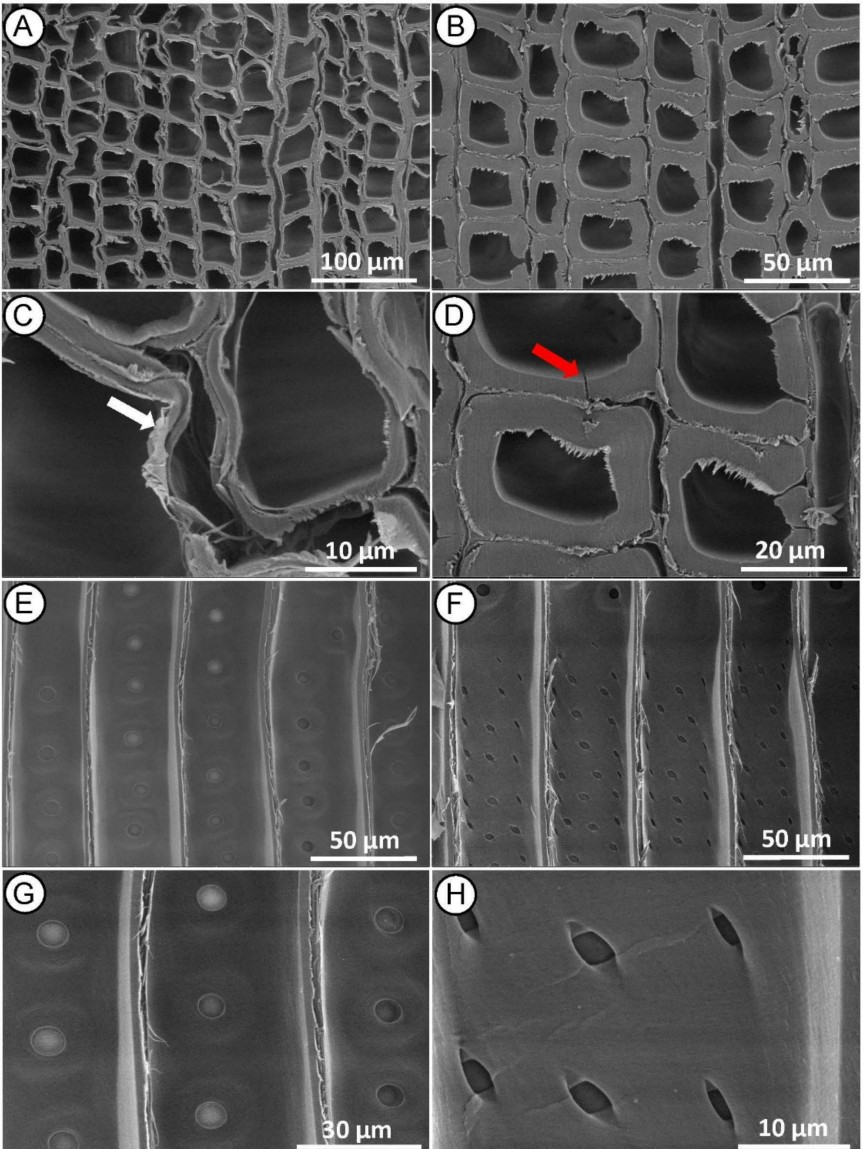

**Figure 4.** SEM micrographs of Norway spruce heartwood after MI MW treatment. Cross-section showing collapsed tracheids in earlywood ((**A**,**C**)—white arrow) and cracked tracheids in latewood ((**B**,**D**)—red arrow). Radial-section with mostly aspirated bordered pits on the tracheid walls (**E**,**G**) and mainly undamaged piceoid cross-field pits (**F**,**H**).

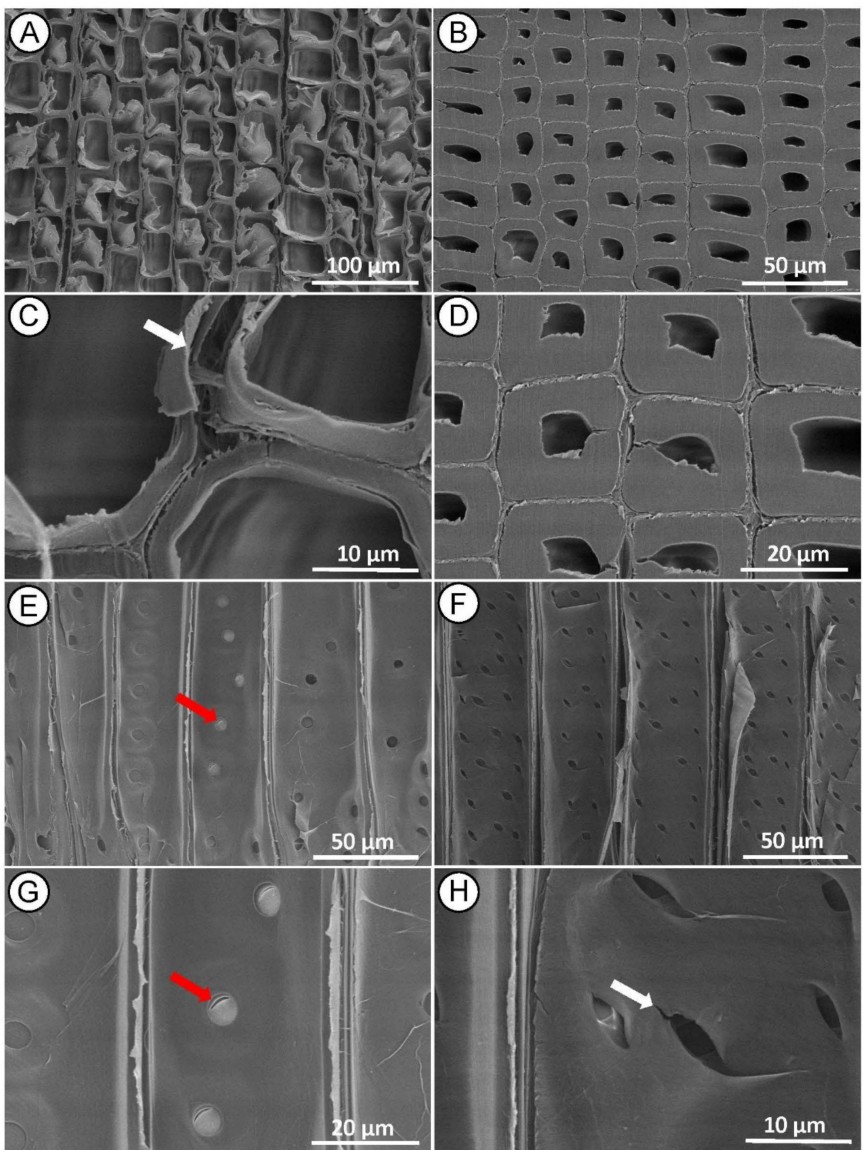

**Figure 5.** SEM micrographs of Norway spruce sapwood after MI MW treatment. Cross section showing collapsed tracheids in earlywood ((**A,C**)—white arrow) and mainly normal tracheids structure in latewood (**B,D**). Radial section showing bordered pits with damage to the pit membrane between the tracheid walls ((**E,G**)—red arrows) and evident cracks in piceoid cross-field pits poles ((**F,H**)—white arrow).

### 3.4. High-Intensity Microwave Treated Heartwood and Sapwood

The heartwood samples modified with HI MW treatment appeared to be the most damaged in microstructure, supporting the results of decreasing skeletal density. In cross-section, we observed tissue distortion and collapsed tracheids in the earlywood (Figure 6A,C) and tracheids detachment in the middle lamella region of the latewood (Figure 6B,D). The greatest damage was observed in the radial section, where we found fractured tracheid (Figure 6E,F), rupture of bordered pits separated from the tracheid wall, and damage of the pit membrane (Figure 6E,G). Cracks (between 8 and 12 μm wide) were also observed at the poles of elliptical pits that propagated along the direction of the microfibril orientation in the cell wall containing cross-field pits (Figure 6F,H) and continued into the S2 cell wall layer. Such significant changes in the anatomical structures, corresponding to HI MW treatment, are consistent with the results reported by other researchers [31]. The cracks are probably result of high-pressure gradient of steam generated within the wood cells

during the intensive treatment, as previously observed by Zhang and co-authors [44,45]. During the HI MW treatment, the cell walls near the pits were severely damaged and cracks make tracheids more susceptible to fractures. We assume that such changes in the anatomical structure makes wood more permeable and thus the penetration of the preservative becomes more effective in impregnation treatments. On the other hand, in this case, the HI MW treatment damaged the wood structure, even at S2 layer level, to such an extent that probably it degrades the mechanical properties of wood.

In the sapwood samples, the earlywood showed a change in the compound middle lamella between adjacent tracheids in cross-section, but without deformation of the tissue as in the previous case (Figure 7A,C). In the latewood, cracks appeared along the radial and tangential walls of the tracheids in the areas of the middle lamella/primary wall regions (Figure 7B,D). In the radial section, we observed wavy longitudinal tracheids that were less fractured compared to the heartwood (Figure 7E,F). These results are in agreement with research [16] reporting that after MW treatment, the bending strength of the sapwood is less affected than of the heartwood in Radiata pine.

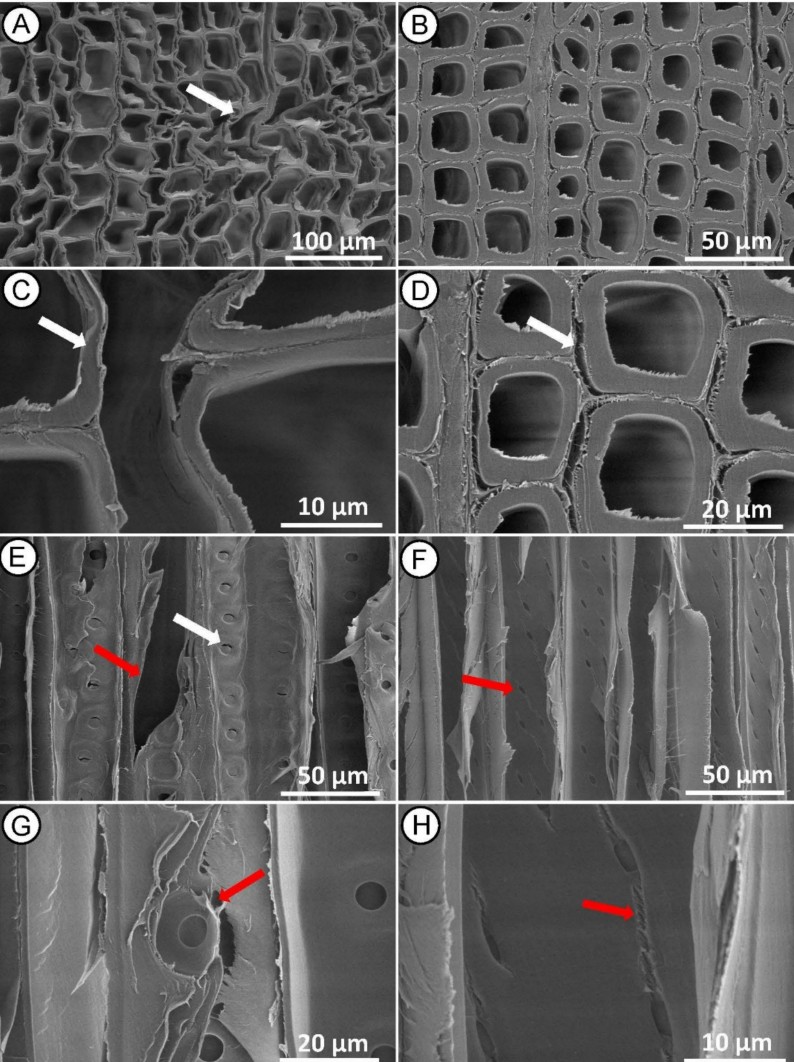

**Figure 6.** SEM micrographs of Norway spruce heartwood after HI MW treatment. Cross section showing tissue distortion ((**A**)—white arrow) and collapsed tracheids in earlywood ((**C**)—white arrow) and detachment of middle lamellas between latewood tracheids ((**B,D**)—white arrow). Radial section showing longitudinal tracheid fractured ((**E**)—red arrow), rupture to bordered pits ((**E**)—white arrow) and damage of pit membrane between tracheid walls ((**G**)—red arrow) and propagation of cracks from the poles of cross-field pits ((**F,H**)—red arrows).

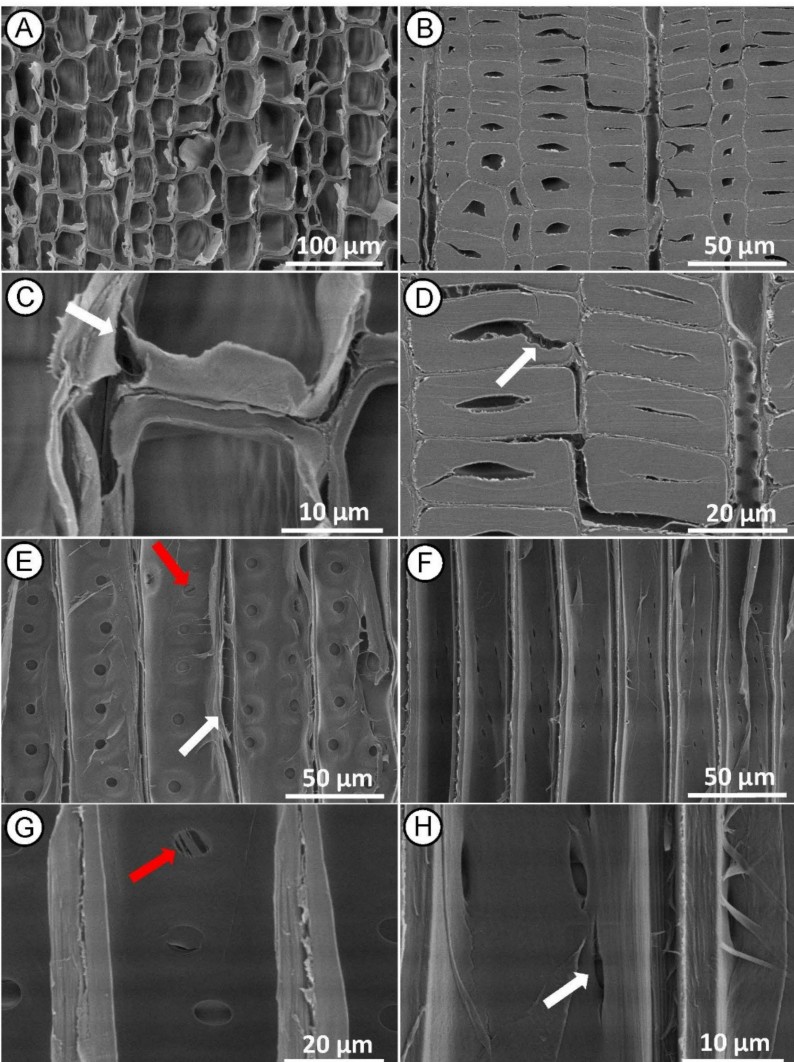

**Figure 7.** SEM micrographs of Norway spruce sapwood after HI MW treatment. Cross section showing collapsed tracheids in earlywood ((**A**,**C**)—white arrow) and partly damages tracheids in latewood ((**B**,**D**)—white arrow). Radial section showing deformation in longitudinal tracheids ((**E**)—white arrow), bordered pits with damage to the pit membrane on the tracheid walls ((**E**,**G**)—red arrows) and crakes in cross-field pits propagated from the poles ((**F**,**H**)—white arrow).

## 4. Conclusions

Scanning electron microscopy analysis of microstructural changes in Norway spruce heartwood and sapwood after MW treatments of different energy intensities provided new insights that are important in the selection of parameters for optimizing the microwave process. The MW treatment resulted in evident changes in wood microstructure in both cross- and radial sections. Here we specifically studied MW treatment for heartwood and sapwood of Norway spruce and found the impact is more prominent and pronounced on heartwood of spruce than on sapwood which was hinted by Torgovnikov and Vinden [16]. Our results revealed that heartwood and sapwood were affected differently by the same MW treatment. Indeed, the heartwood proved to be more affected in its microstructure modification by a HI MW treatment than the sapwood. This fact can also be observed from the changes in absolute and skeletal density. The degree of modification corresponded to the intensity level of the MW treatment. HI produced the highest degree of rupture in the wood microstructure, which led to an increase in wood porosity, and we assume consequently to a higher permeability of the wood, which may facilitate the penetration of the preservative during the impregnation treatment. On the other hand, severe structural damage probably

also significantly reduces the mechanical properties of the wood. Impregnability and mechanical properties are topics for further examination on MW treated material.

We can conclude that to increase the permeability of the sapwood, both moderate and high intensity MW treatment can be applied without causing severe damage to the structure, while in the case of the heartwood, the high intensity treatment should be avoided if we want to preserve its structural integrity and probably also mechanical properties.

These findings on MW treatment may help to implement this method in timber processing to improve the impregnation treatment of wood species with low permeability, allowing the use of local wood resources. When applying this method in practice, additional tests of mechanical properties need to be considered for individual wood species and for different MW treatment intensities.

**Author Contributions:** Conceptualization, M.M., S.G., A.B., D.K., M.P.; methodology, M.M., S.G., A.B., J.Ž., D.K., M.P., S.T.; validation, M.M., S.G.; formal analysis, A.B., D.K.; investigation, M.M., S.G., A.B., D.K.; data curation, M.M., A.B., D.K.; writing—original draft preparation, S.G., A.B., J.Ž.; writing—review and editing, M.M., S.G., A.B., J.Ž., M.P., D.K.; visualization, M.M., S.G.; supervision, M.M., S.T., J.Ž., M.P.; project administration, M.P., S.T.; funding acquisition, S.G., M.P., S.T. All authors have read and agreed to the published version of the manuscript.

**Funding:** The research was supported by the Program P4-0015, co-financed by the Slovenian Research Agency and additionally supported by the CMEPIUS, Slovenia, and UCOST, Dehradun, India.

**Institutional Review Board Statement:** Not applicable.

**Informed Consent Statement:** Not applicable.

**Data Availability Statement:** The datasets in this study are available within this article. The data which were derived from the original datasets but not aforementioned are available upon requests.

**Acknowledgments:** The authors wish to thank Samo Grbec for his immense help with sample preparation and José Gonçalves for his contribution with statistical analysis. S.G is thankful to the Director, FRI, India, the Registrar, FRIDU, Dehradun, India and the Dean (A), FRIDU, Dehradun for their motivation and support during the study period.

**Conflicts of Interest:** The authors declare no conflict of interest.

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
