# Peer review of "Effects of Different Energy Intensities of Microwave Treatment on Heartwood and Sapwood Microstructures in Norway Spruce"

_forests, doi:10.3390/f12050598_

Round 1
Reviewer 1 Report
The manuscript is informative and has got academic as well as practical significance. However, some minor suggestions have been provided and may be addressed.
- The importance of microwave treatment on Norway spruce wood may be mentioned in the introduction.
- Samples were conditioned to 11 percent EMC before microwaving. Why such low moisture content was selected when the purpose was to improve the permeability of wood through increased steam/vapour pressure by MW heating.(Line 143)
- If possible, the duration of irradiation, MW power used and moisture reduction due to microwave treatment may be provided in section 2.2.
- The equations for calculating percentage porosity and total pore volume from skeletal and envelope volume may be provided (LIne 184)
- Are the differences in densities and porosity in statistically significant? (Section 3.1)
- Please check the statement in line 237 " --- wide sap wood rings in these samples". If the samples are from heartwood, how sapwood rings present in that.
- Check the Figure numbers in the paragraphs represented between line 274 - 280 and Line 294-298
-
The phrase in line no. 366 may be corrected.
Some additional minor corrections have also been included in the attached pdf copy of the manuscript.

Author Response
Thank you for your time and effort. Please find our response in attached word document.
Maks Merela

Reviewer 2 Report
The manuscript is focused on the evaluation of the changes in the anatomical structure of Norway spruce (Picea abies (L.) Karst.) heartwood and sapwood after microwave modification in order to develop most effective treatment in terms of applied energy without causing significant structural damage. Analysis with light and scanning electron microscopy were performed in order to evaluate the effect of microwave treatment for two different energy intensities, moderate and high intensity.
Overall, the manuscript is well-written, the experimental data are presented and discussed in a clear and concise manner. The references included and referred to experimental design are appropriate. Some minor revision is needed. My comments are included in the attached PDF.

Author Response

(The authors gave the same response as above.)

Reviewer 3 Report
The manuscript “Effects of different energy intensities of microwave treatment on heartwood and sapwood microstructures in Norway spruce” offers some repetition of known results on the effect of MW on wood structure. The objectives of the study “to develop most effective treatment..…without causing significant structural damage” (row 27-28) are too ambitious having in mind the used equipment but also contradict the aims of MW irradiation in the majority of its applications on wood. Some moments in the manuscript that can be improved are:
- Fundamental question: why is wood conditioned (20°C and 60%RH) to 11% before MW irradiation? Improvement of permeability is needed for the raw material (spruce wood), which, once dried, becomes less permeable. MW at the selected frequency affect only water, which is bounded and not so “movable”.
- Fig. 1 doubles the results in Table 1 and should be omitted.
- Are the differences in Table 1 statistically different? Statistics?
- If “yes” – why does the skeletal density decreases? The manuscript needs some discussion on the density changes, which are only reported at the moment.
- Fig. 2 is not of great use and should be omitted. “…the higher proportion of latewood in MI MW samples” (row 236) is very debatable as well as the effect of sample shape on the measurements.
- Rows 264-269 brings no relevant information. The usefulness of Fig 3 should be questioned since both LM and SEM pictures are at low magnification and do not bring some information. Examples are the wood features described in rows 278-280; it is impossible to see them in Fig. 3, some of them cannot appear in transversal section. Omit the LM.
- Rows 317-318 – the authors cannot be categorical on the effect of MW on the mechanical properties since they have not measured them.
Author Response

(The authors gave the same response as above.)
